# Genetically diverse mouse models of SARS-CoV-2 infection reproduce clinical variation in type I interferon and cytokine responses in COVID-19

Shelly J. Robertson[1,2], Olivia Bedard[3], Kristin L. McNally[1], Carl Shaia[4], Chad S. Clancy[4], Matthew Lewis[1], Rebecca M. Broeckel[1], Abhilash I. Chiramel[1], Jeffrey G. Shannon[2], Gail L. Sturdevant[1,2], Rebecca Rosenke[4], Sarah L. Anzick[5], Elvira Forte[3,8], Christoph Preuss[3], Candice N. Baker[3], Jeffrey M. Harder[3], Catherine Brunton[3], Steven Munger[3], Daniel P. Bruno[5], Justin B. Lack[5], Jacqueline M. Leung[5], Amirhossein Shamsaddini[5], Paul Gardina[5], Daniel E. Sturdevant[5], Jian Sun[5], Craig Martens[5], Steven M. Holland[6], Nadia A. Rosenthal[3,7] ✉ & Sonja M. Best[1,2] ✉

Inflammation in response to severe acute respiratory syndrome coronavirus-2 (SARS-CoV-2) infection drives severity of coronavirus disease 2019 (COVID-19) and is influenced by host genetics. To understand mechanisms of inflammation, animal models that reflect genetic diversity and clinical outcomes observed in humans are needed. We report a mouse panel comprising the genetically diverse Collaborative Cross (CC) founder strains crossed to human ACE2 transgenic mice (K18-hACE2) that confers susceptibility to SARS-CoV-2. Infection of CC x K18-hACE2 resulted in a spectrum of survival, viral replication kinetics, and immune profiles. Importantly, in contrast to the K18-hACE2 model, early type I interferon (IFN-I) and regulated proinflammatory responses were required for control of SARS-CoV-2 replication in PWK x K18-hACE2 mice that were highly resistant to disease. Thus, virus dynamics and inflammation observed in COVID-19 can be modeled in diverse mouse strains that provide a genetically tractable platform for understanding anti-coronavirus immunity.

The COVID-19 pandemic continues to pose a global threat to public health, due in part to the sequential evolution of new variants of concern (VOC) selected to evade immunity generated from previous infection or vaccination. The extreme variability in patient responses to infection, ranging from asymptomatic to life-threatening illness[1–3],

remains only partially understood. In addition, persistent post-COVID health problems can be severe and include a significant risk for future morbidity and mortality[4]. The variability in disease presentation is dependent on genetic polymorphisms, age, sex, and the presence of underlying conditions[1,5,6], underscoring the need for research models

[1]Laboratory of Virology, Rocky Mountain Laboratories, National Institute of Allergy and Infectious Diseases, NIH, Hamilton, MT, USA. [2]Laboratory of Neurological Infections and Immunity, Rocky Mountain Laboratories, National Institute of Allergy and Infectious Diseases, NIH, Hamilton, MT, USA. [3]The Jackson Laboratory, Bar Harbor, ME, USA. [4]Rocky Mountain Veterinary Branch, Rocky Mountain Laboratories, National Institute of Allergy and Infectious Diseases, NIH, Hamilton, MT, USA. [5]Research Technologies Branch, National Institute of Allergy and Infectious Diseases, NIH, Bethesda, MD, USA. [6]Division of Intramural Research, National Institute of Allergy and Infectious Diseases, NIH, Bethesda, MD, USA. [7]National Heart and Lung Institute, Imperial College London, London, UK. [8]Present address: Springer Nature, New York, NY, USA. ✉e-mail: Nadia.Rosenthal@jax.com; sbest@niaid.nih.gov

that reflect the diverse biology and pathology of SARS-CoV-2 infection in order to provide optimal platforms for preclinical development of novel therapeutic strategies.

Type I interferon (IFN-I) is essential for the control of virus replication, but its functions in COVID-19 are complex and poorly understood, with evidence for roles in both protective and pathogenic host response[7]. The success of IFN-I in controlling virus replication and orchestrating an effective inflammatory response may relate to the timing of IFN-I induction relative to peak SARS-CoV-2 replication, the intensity of IFN-I expression and the kinetics of response resolution[8]. In humans, favorable outcomes of COVID-19 have been associated with robust early IFN-I responses that are subsequently resolved along with other disease signs[9]. However, the more severe manifestations of COVID-19, including reduced oxygen saturation, higher inflammatory responses, and lower anti-viral antibody responses, have been observed in patients with either an early IFN-I response that fails to resolve[10], or an IFN-I response that is substantially delayed relative to peak virus replication[11-14]. The roles of IFN-I in a pathogenic response may be linked to failure to control virus replication. However, late, or sustained IFN-I responses may drive pathogenesis through failure to orchestrate an effective adaptive response, exacerbation of inflammatory monocyte responses, and inhibition of tissue repair[7,10]. In mice, loss of IFN-I signaling reduces inflammatory cell recruitment in the lung, but a direct antiviral role of IFN-I is less clear[15,16]. Thus, additional experimental models are needed that inform how the timing, magnitude, and duration of innate immunity relative to virus replication dynamics determines virus dissemination and whether inflammatory responses are protective or pathogenic.

Current mouse models of COVID-19 are important tools in understanding drivers of inflammatory responses and pathology following infection with SARS-CoV-2, but are typically generated on invariant inbred backgrounds, poorly reflecting the diversity of patient outcomes and the impact of host genetics on SARS-CoV-2 infection and response to treatment. Mice humanized for the main cellular receptor for SARS-CoV-2 entry, angiotensin-converting enzyme 2 (ACE2)[17], have been essential pre-clinical infection models[18]. The most widely used model carrying a human ACE2 gene (K18-hACE2) supports high early virus replication in lung epithelial cells, resulting in inflammatory cell infiltration, interstitial edema, and focal consolidation of the lung with rapid lethality following virus dissemination to the central nervous system (CNS)[19-24]. However, this model on a standard C57BL/6 J genetic background does not reflect the wide variety of COVID-19 outcomes in humans[1,5,6]. Indeed, in the absence of comparative models with different outcomes, it is not known what human response is being modeled by SARS-CoV-2 infection of K18-hACE2 mice specifically regarding how IFN-I response dynamics relates to control of virus replication and pathology.

To determine if incorporating genetic diversity could recapitulate the broad phenotypic variation observed in human COVID-19, we crossed the K18-hACE2 strain to the genetically distinct inbred strains A/J, 129S1/SvImJ, NOD/ShiLtJ, NZO/HILtJ, PWK/PhJ, CAST/EiJ, and WSB/EiJ that along with C57BL/6 J comprise the CC founders. Together, these strains represent 90% of the genetic diversity in *M. musculus* populations[25,26], and therefore can be used to understand the genetic regulation of complex immune responses. Notably, NOD/ShiLtJ and NZO/HILtJ mice are commonly used in studies of diabetes and metabolic dysfunction, both factors contributing to COVID-19 severity[27]. In addition to the 8 CC founder strains, BALB/cJ and DBA/2 J strains were included because BALB/cJ is widely used in infectious disease modelling, and DBA/2 J is a founder strain for BXD strains also used in genetic mapping[28]. Here, we show that by using these genetically diverse mice we can model both early and delayed innate immune responses associated with differing control of virus replication kinetics and pathology in the lung, and formally demonstrate an effective antiviral role for IFN-I if produced early. These mice, therefore, represent invaluable tools to understand how IFN-I and proinflammatory responses are orchestrated to control SARS-CoV-2 replication and pathology.

## Results

### Diverse outcomes of SARS-CoV-2 infection in CC x K18-hACE2 F1 mice

F1 progeny of CC founder x K18-hACE2 mice (hereafter abbreviated CC x K18-hACE2) were infected via intranasal inoculation with $10^3$ plaque-forming units (pfu) SARS-CoV-2 strain nCoV-WA1-2020, MN985325.1 and mice were monitored for 21 days. This virus dose is sub-uniformly lethal in the prototype K18-hACE2 strain (C57BL/6 J background), resulting in 80-90% of mice reaching end-point criteria by 7 days post-infection (dpi) (Fig. 1a). Infection was confirmed in all survivors by seroconversion to SARS-CoV-2 nucleoprotein. Comparative analysis of the eight CC x K18-hACE2 F1 cohorts documented a broad spectrum of weight loss and survival curves characteristic to each CC founder genotype, some of which were sex-specific. Highly sensitive strains included C57BL/6 J and A/J x K18-hACE2 that lost approximately 20% of their starting weight between 4 and 7 dpi with no clear sex bias (Fig. 1a). In contrast, CC x K18-hACE2 from PWK, NZO, 129S1/J (Fig. 1b), BALB/cJ and DBA/2 J (Fig. S1a, b) were comparatively resistant to clinical disease, generally losing 5-10% starting weight with ~80% of mice surviving infection. Finally, CC x K18-hACE2 of three strains (CAST, NOD, WSB) had marked sexual dimorphism, with CAST and NOD males susceptible to lethal disease, although differences in weight loss between sexes were not apparent (Fig. 1c). Yet another phenotype was observed in WSB x K18-hACE2, where the infection was uniformly lethal in females. Weight loss in males began at 6 dpi, later than other groups, and surviving males exhibited sustained weight loss in contrast to the rapid weight gain associated with recovery of other CC x K18-hACE2 cohorts (Fig. 1c).

### Virus replication and dissemination in CC x K18-hACE2 mice

In K18-hACE2 mice, SARS-CoV-2 titers peak in the lung at 2-3 dpi, and high titers of the virus can be isolated from the CNS which may contribute to lethality in this model[17,29]. Thus, virus replication kinetics were determined at 3 and 6 dpi in the lung and brain of CC x K18-hACE2 cohorts (Fig. 2a, Fig. S1c). Sensitive founder strains K18-hACE2 and A/J x K18-hACE2 showed high levels of infectious SARS-CoV-2 at 3 dpi in lung homogenates that was reduced by 1-2 $\log_{10}$ by 6 dpi with no differences between sexes. At 3 dpi, infectious virus was generally not recovered from the CNS, but ~50% of K18-hACE2 and A/J x K18-hACE2 had high virus burden in the CNS by 6 dpi ($10^6$-$10^8$ pfu/g tissue) suggesting significant virus dissemination outside of the lung. In contrast, peak lung virus titer in the most resistant PWK x K18-hACE2 was 150-fold lower than the sensitive strains at 3 dpi and was below the limit of detection at 6 dpi. Control of virus replication in PWK x K18-hACE2 was also evident in the CNS where detection of infectious virus was sporadic. Interestingly, additional CC x K18-hACE2 classed as resistant had equivalent peak viral titers in the lung to the sensitive K18-hACE2 and A/J x K18-hACE2, but they tended to control replication by 6 dpi to levels 1 $\log_{10}$ lower than sensitive mice and had less virus burden in the CNS. Finally, lung titers in (CAST, NOD, and WSB) x K18-hACE2 were not different between males and females despite differences in clinical severity associated with sex. Relative expression of the *K18-hACE2* transgene in the lung was not different between genetic backgrounds (Fig. S2b) and ACE2 distribution was not distinguishable between sensitive (K18-hACE2) and resistant (PWK x K18-hACE2) mice by immunohistochemistry (Fig. S2c). Thus, CC x K18-hACE2 can be further stratified into a) sensitive with high sustained virus replication in lung and CNS (C57BL/6 J, A/J), b) resistant with lower peak virus titer or earlier control of replication in the lung and no or low dissemination to other organs (PWK, NZO, 129S1/J), and c) sex-biased

 

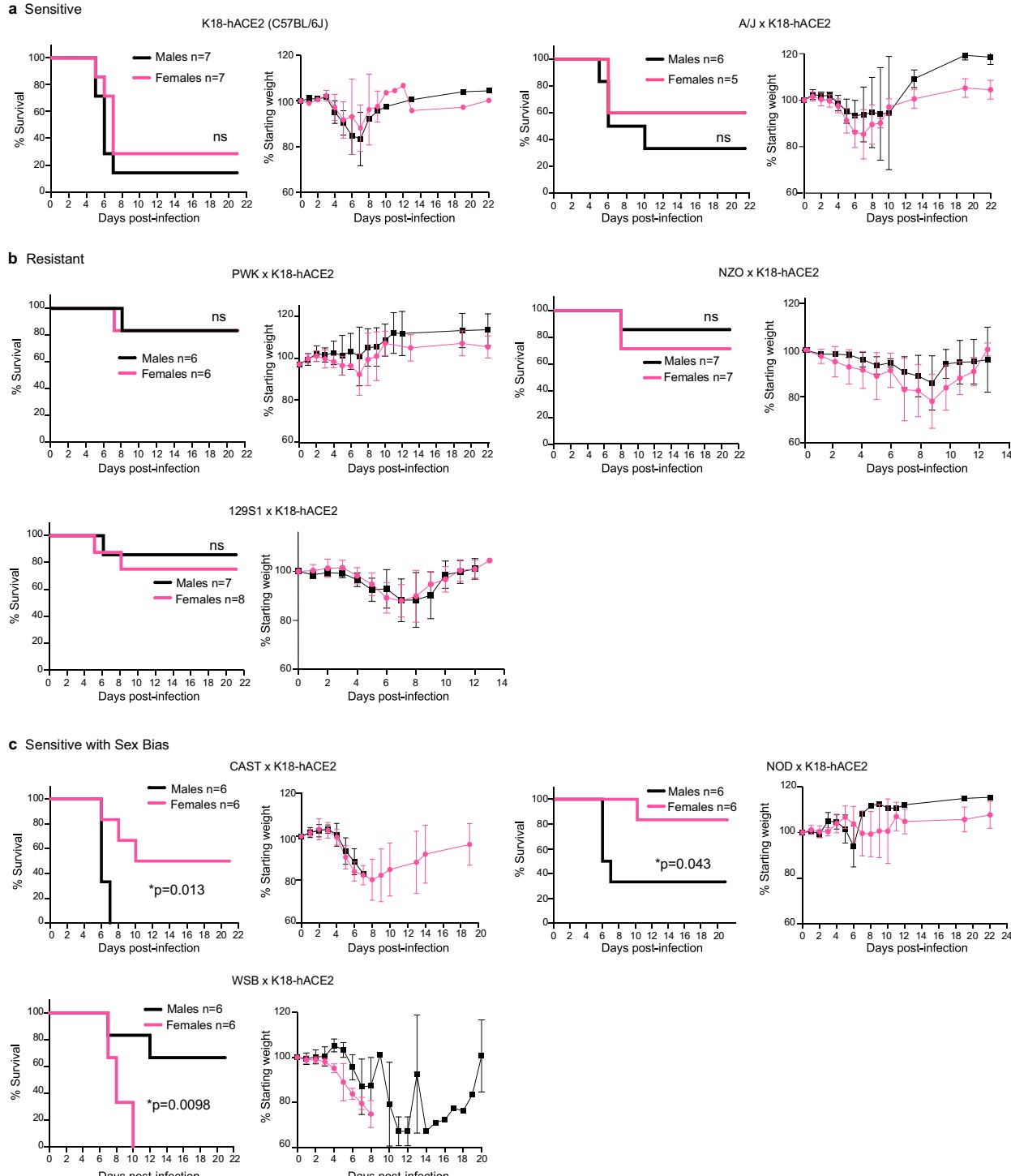

**Fig. 1 | CC x K18-hACE2 mice demonstrate a range of clinical severity following infection with SARS-CoV-2.** Mice were infected intranasally with $10^3$ pfu of SARS-CoV-2 and monitored for survival and weight loss. CC x K18-hACE2 were classified as: **a** sensitive with 50% or greater mortality and no difference between males and females, **b** resistant with 35% or less mortality, or **(c)** sensitive with a sex bias as having statistically significant difference between male and female mice. Graphs of percent starting weight show the mean ± SD. Biological replicates were examined over at least 2 independent experiments and the numbers of mice assessed per strain (male: female) were as follows: (K18-hACE2 (7:7); A/J x K18-hACE2 (6:5); PWK x K18-hACE2 (6:6); NZO x K18-hACE2 (7:7); 129S1 x K18-hACE2 (7:8); CAST x K18-hACE2 (6:6); NOD x K18-hACE2 (6:6); WSB x K18-hACE2 (6:6). The Mantel-Cox log-rank test with 95% confidence interval was used to compare male versus female survival curves for each strain. *$p < 0.05$ was considered statistically significant.

outcomes in survival independent of virus titer in the lung suggesting a sex-based difference in host response (CAST, NOD, WSB). Together, these data demonstrate that major differences in the dynamics of SARS-CoV-2 replication and host sensitivity to disease observed in humans can be modeled through the use of genetically diverse mouse strains (Summarized in Table S1).

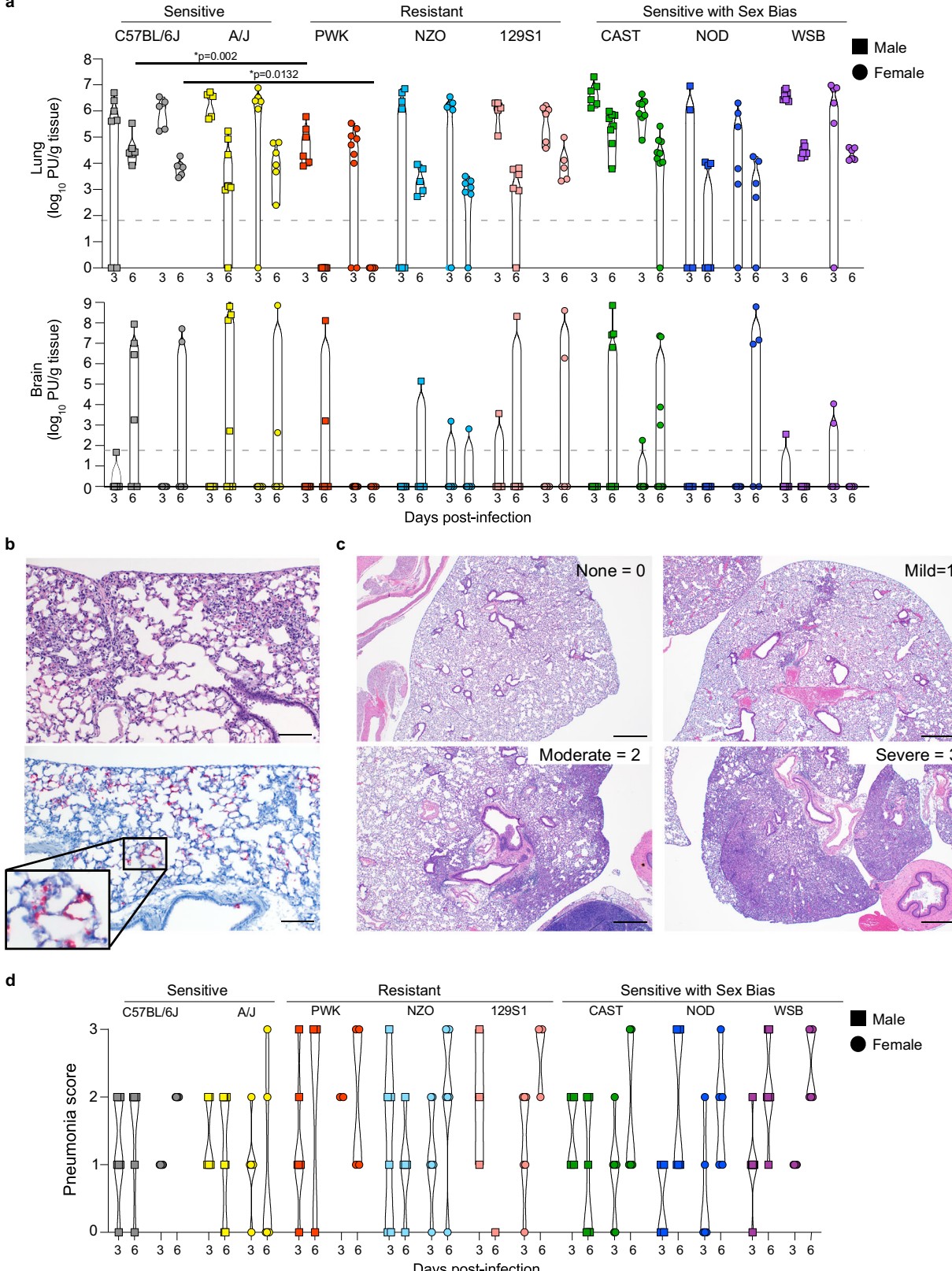

## Histopathology in lung and brain from SARS-CoV-2-infected CC x K18-hACE2

The general pathological changes in SARS-CoV-2-infected K18-hACE2 mice were similar to those previously described[19,23,24,30] although varied in severity (Fig. 2b–d). Inflammatory infiltrates evident by 3 dpi included perivascular lymphocytes with alveolar septa thickened by neutrophils, macrophages, and edema. At 6 dpi, pulmonary pathology was multifocal, and consistent with interstitial pneumonia including type II pneumocyte hyperplasia, septal, alveolar and perivascular inflammation comprised of lymphocytes, macrophages and neutrophils, with alveolar fibrin and edema evident. Bronchiolar pathology was not observed in these mice. Lung pathology was classified as none,

**Fig. 2 | Virus replication kinetics and pathology in CC x K18-hACE2. a** Violin plots of virus titers (pfu/g tissue) in lung and brain of male (squares) and female (circles) mice measured at 3 and 6 dpi. **b** H&E staining (upper) and in-situ hybridization for SARS-CoV-2 nucleic acid (red) predominantly localizing to pneumocytes in areas of lung inflammation (lower) using serial sections. Scale bars represent 100 μm. Images are representative of lung sections from all CC x K18-hACE2 mice at 3 dpi. **c** Examples of scoring used to evaluate lung pathology in all biological replicates of CC x K18-hACE2 mice at 3 and 6 dpi. Scale bars represent 500 μm. **d** Violin plots of lung pathology scores for CC x K18-hACE2 males and females at 3 and 6 dpi. Source data are provided as a Source Data file. The numbers of mice assessed per strain (male: female) were as follows: 3 dpi (K18-hACE2 (5:7); A/J x K18-hACE2 (5:5); PWK x K18-hACE2 (7:6); NZO x K18-hACE2 (7:7); 129S1 x K18-hACE2 (6:5); CAST x K18-hACE2 (8:6); NOD x K18-hACE2 (5:4); WSB x K18-hACE2 (6:9), and 6 dpi (K18-hACE2 (6:6); A/J x K18-hACE2 (6:7); PWK x K18-hACE2 (6:7); NZO x K18-hACE2 (7:5); 129S1 x K18-hACE2 (5:5); CAST x K18-hACE2 (8:9); NOD x K18-hACE2 (4:6); WSB x K18-hACE2 (5:7). Kruskal-Wallis test with Dunn's posttest was used to compare lung virus titers in each strain/sex to that of K18-hACE2 (males and females separated). *$p < 0.05$ was considered statistically significant.

mild (rare scattered inflammatory foci), moderate (coalescing inflammatory foci) or severe (widespread, large inflammatory foci) as exemplified in Fig. 2c. Surprisingly, most resistant strains and those with sex bias tended to have higher pneumonia scores in lungs compared to the sensitive K18-hACE2 and A/J x K18-hACE2 (Fig. 2d). Lung distribution of viral RNA was limited to type I and II pneumocytes in all strains (Fig. 2b). Importantly, these data suggest that lung pathology can be exacerbated in the context of effective control of SARS-CoV-2 replication but still be associated with rapid recovery, as modeled in PWK x K18-hACE2 and NZO x K18-hACE2. In contrast, higher pathology scores associated with uncontrolled virus replication in the lung and higher sensitivity to severe disease can be modeled by in WSB x K18-hACE2.

CNS pathology was scored as present or absent, as it generally consisted of subtle inflammatory foci, including perivascular cuffing and gliosis associated with necrotic cells observed in K18-hACE2 and DBA x K18-hACE2 males (Fig. S3a–c, g). However, pathology in the CNS of CAST x K18-hACE2 was striking in that microthrombi were evident in capillaries with extensive hemorrhage in the absence of encephalitis. Although microthrombi were associated with viral RNA distribution in the same area (Fig. S3d–f), microhemorrhage was not observed as extensively in infected K18-hACE2 mice. In addition, infection with $10^3$ pfu SARS-CoV-2 was uniformly lethal in male CAST x K18-hACE2, despite these mice having some of the lowest levels of pathology in the lung at 6 dpi of all mice examined. Thus, CAST x K18-hACE2 mice may be a suitable model for the examination of microthrombus formation sometimes observed in severe human COVID-19[31–33].

### Distinct timing and amplitude of cytokine and chemokine induction in CC x K18-hACE2 mice

In-depth longitudinal analyses of immune responses in patients with COVID-19 have defined immunological correlates of disease outcome[8,10,12,34–44]. In addition to high early IFN responses, moderate COVID-19 is associated with increased plasma levels of a core of proinflammatory cytokines, chemokines and growth factors (e.g., IL-1a, IL-1β, IFNα, IL-12p70, and IL-17A) that are effectively resolved during infection[10]. In contrast, severe COVID-19 includes sustained expression of these markers with additional signatures including IL-6, IL-10, IL-18, IL-23, TNF, and eotaxin among others suggesting that specific timing and failure to resolve inflammatory responses are important factors in disease progression[10].

In SARS-CoV-2-infected CC x K18-hACE2, cytokines and chemokines were quantified in the serum and bronchoalveolar lavage fluid (BAL) at 3 and 6 dpi. Cytokine levels were much higher in BAL than in serum (Fig. S4a–c; individual data points are provided in Figs. S5–7). IFNα was variably induced at 3 dpi in all crosses with K18-hACE2 (Fig. 3a and Fig. S1e). Strikingly, K18-hACE2 and CC x K18-hACE2 of A/J, CAST, and NOD produced relatively low IFNα compared to PWK, NZO, 129S1/J, and WSB that produced 2- to 10-fold higher levels. At 3 dpi, IFNα production was similar in males and females within each strain except for PWK x K18-hACE2 males produced significantly higher levels than females, and in DBA x K18-hACE2 where females were higher (Fig. S1d). Importantly, high early IFNα expression at 3 dpi was resolved by 6 dpi in resistant ((PWK, NZO and 129S1/J) x K18-hACE2) that most effectively controlled virus replication in the lung and CNS. In comparison, low

levels of IFNα were observed in K18-hACE2 mice at 3 and 6 dpi despite high early virus burden in the lung and failure to control disseminating infection by 6 dpi (Fig. 3b, c; Fig. S5). Thus, PWK x K18-hACE2 may represent a model of effective induction and resolution of IFN responses associated with viral control. A third distinct phenotype was observed in WSB x K18-hACE2 that had high IFNα expression in BAL at 3 dpi equivalent to PWK x K18-hACE2, but these mice did not control virus replication in the lung as effectively as PWK F1 mice and had sustained weight loss in surviving individuals. Indeed, WSB x K18-hACE2 produced some of the highest IFNα among all genetic backgrounds at 3 dpi (Fig. 3a) along with high levels of cytokines and chemokines (Fig. S4a–c), which may be consistent with significant weight loss observed in both sexes (Fig. 1c). In addition, WSB x K18-hACE2 had equivalent lung virus titers to K18-hACE2 mice at 3 and 6 dpi, but virus was only sporadically recovered from brain (Fig. 2a). Despite similarities in virus burden between sexes, female WSB x K18-hACE2 mice were significantly more sensitive to severe clinical disease (Fig. 1c) and lung pathology (Fig. 2d). Thus, the WSB strain background may model the situation where high IFN responses are not associated with viral control and contribute to pathogenesis.

To further compare the relative expression of immune effectors in BAL, unsupervised hierarchical cluster analysis was performed on data from sensitive (K18-hACE2 and A/J x K18-hACE2) and three resistant strains that did not show a sex bias ((PWK, NZO, 129S1/J) x K18-hACE2). SARS-CoV-2-infected mice were grouped into four general clusters each with distinct profiles of cytokine and chemokine expression. As shown in Fig. 3d, Cluster 1 was comprised of resistant mice at 3 dpi that expressed proinflammatory cytokine IL-6, TH1 cytokines (IL-12p70, IFNγ, IL-27), inflammasome associated cytokine (IL-1beta and IL-18) and chemokines (CXCL10, Gro-α/KC, CCL2, CCL3, CCL4, CCL5, CCL7 and CXCL2) (designated as Group A). At 6 dpi, some Group A cytokines were resolving (IL-6, IL-27, CCL2 and CXCL10) (Fig. 3e; Table S2; Table S3) while IL-2, IL-4, IL-5, IL-13, IL-18, IL-1b, GM-CSF, eotaxin, TNF-α, IFN-γ, and IL10 (designated as Group B) were added to the cytokine signature in resistant mice (Fig. 3d, Cluster 2). Interestingly, resistant PWK and NZO x K18-hACE2 mice generally exhibited a unique profile characterized by tempered levels of Group A and Group B along with high production of Gro-α/KC, CCL7, CXCL10, CCL2, eotaxin, and IFNγ, at 6 dpi, a timepoint coincident with the most efficient clearance of virus from the lung (Cluster 4). In contrast, sensitive strains generally failed to produce Group A cytokines at 3 dpi (Cluster 3) and if induced, production of both Group A and Group B mediators occurred at the later timepoint (Fig. 3d, Cluster 2). In contrast, sensitive CC x K18-hACE2 produced significantly higher levels of IL-10 (immune suppressive cytokine) and CCL4 (monocyte and neutrophil recruiting chemokine) at 6 dpi compared to resistant F1 (Fig. 3e; Table S2).

### Type I IFN signaling controls early SARS-CoV-2 replication and coordinates inflammatory responses in PWK x K18-hACE2

A correlation coefficient analysis was performed to determine the relationship between IFNα and other immune mediators, focusing on sensitive (K18-hACE2) and resistant (PWK x K18-hACE2) mice. In PWK x K18-hACE2, high levels of IFNα at 3 dpi positively correlated with a small subset of cytokines (IFNγ, IL-12p70, IL-6, IL-27, and CCL5) and negatively correlated with IL-10, CXCL10, and CXCL2 (Fig. 3f). In K18-

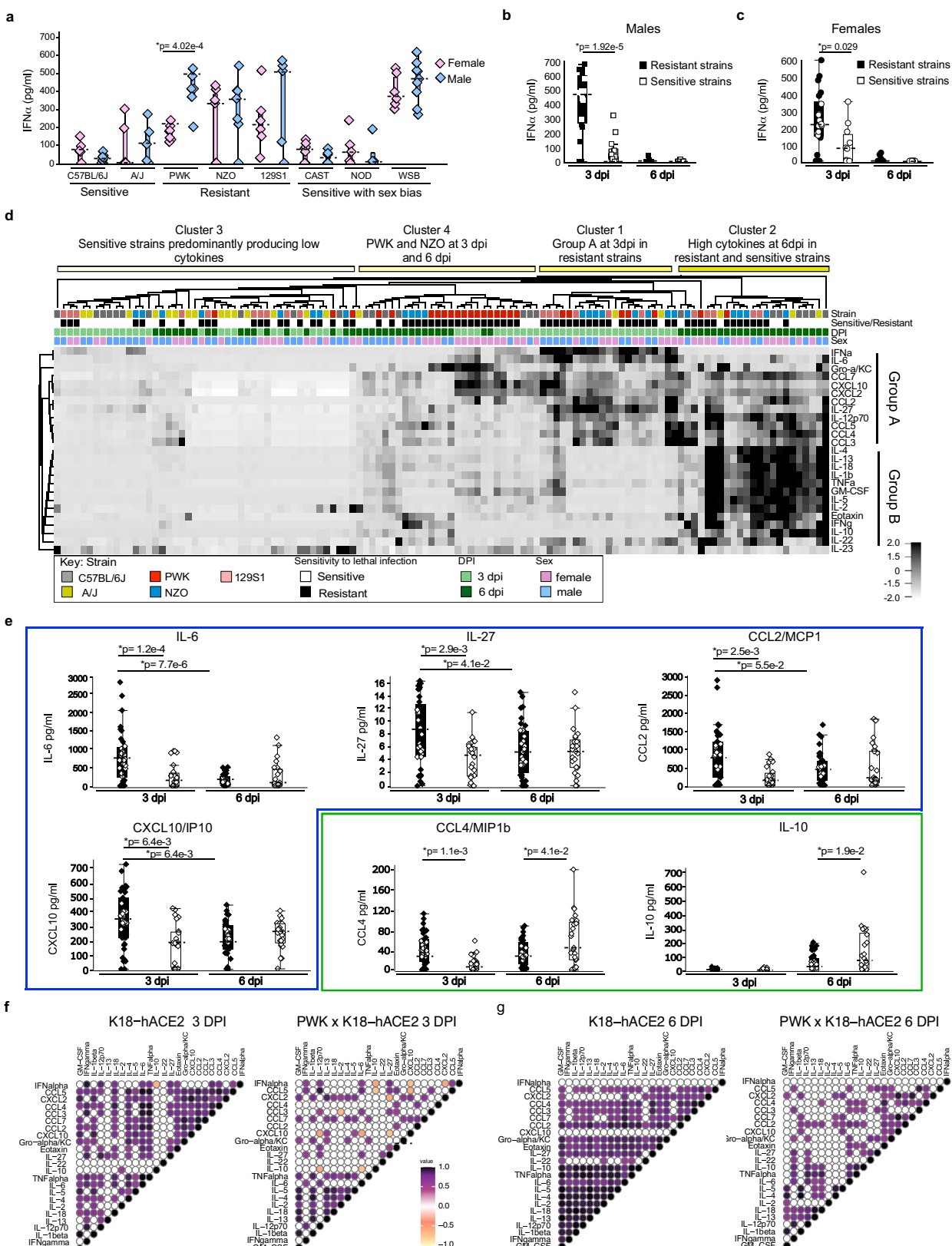

hACE2 mice, low IFNα production correlated with low expression of these same immune mediators along with pro-inflammatory TNFα and IL-18, and immune cell recruiting chemokines (eotaxin, CCL3, and CCL4). Although IFNα expression was resolved to near basal levels in both sensitive and resistant F1 mice at 6 dpi (Fig. 3b, c) expression correlations between immune mediators increased in sensitive

K18-hACE2, but generally decreased in resistant PWK x K18-hACE2 (Fig. 3g), consistent with high dysregulation in K18-hACE2.

To obtain an unbiased, global assessment of host responses in a direct comparison between resistant and sensitive mouse models, we performed Visium spatial transcriptomics on lungs from male K18-hACE2 and PWK x K18-hACE2 at 0 (naïve), 3 and 6 dpi. Confidence

**Fig. 3 | Timing of IFN-I and inflammatory cytokine responses in the lung is associated with clinical severity in CC x K18-hACE2. a** Box plots of IFNα protein in BAL from CC x K18-hACE2 mice at 3 dpi. **b, c** IFNα protein in BAL from (**b**) male and (**c**) female resistant (PWK, NZO and 129S1 x K18-hACE2) and sensitive (K18-hACE2 and A/J x K18-hACE2) mice at 3 and 6 dpi. Biological replicates were examined over at least 2 independent experiments and the numbers of mice assessed per strain (male: female) were as follows: 3 dpi (K18-hACE2 (5:7); A/J x K18-hACE2 (5:5); PWK x K18-hACE2 (7:6); NZO x K18-hACE2 (7:7); 129S1 x K18-hACE2 (6:5); CAST x K18-hACE2 (8:6); NOD x K18-hACE2 (5:4); WSB x K18-hACE2 (6:9), and 6 dpi (K18-hACE2 (6:6); A/J x K18-hACE2 (6:7); PWK x K18-hACE2 (6:7); NZO x K18-hACE2 (7:5); 129S1 x K18-hACE2 (5:5); CAST x K18-hACE2 (8:9); NOD x K18-hACE2 (4:6); WSB x K18-hACE2 (5:7). Two-tailed Student t-test with 95% confidence interval was used to compare cytokine production in (**a**) male versus female for each strain, (**b**) resistant versus sensitive males, and (**c**) resistant versus sensitive females. *$p < 0.05$ was

considered statistically significant. **d** Heatmap of cytokine and chemokine responses in BAL. **e** Box plots of resolving group A cytokines in resistant CC x K18-hACE2 mice (IL-6, IL-27, CCL2, CXCL10), or increasing group B cytokines in sensitive CC x K18-hACE2 mice (IL-10, CCL4). Box plots show data from individual mice (squares, circles, diamonds), the horizontal dashed lines represent the median value, the box indicates the first and third quartiles, and the whiskers indicate the maxima and minima values (Qlucore software). Source data are provided as a Source Data file. Two-tailed, unpaired Student's t-test with Holm-Bonferroni multiple comparison posttest and 95% confidence interval were used to compare cytokine production in resistant versus sensitive strains (males and females combined). *$p < 0.05$ was considered statistically significant. **f, g** Correlation plots of IFNα and cytokine panel for K18-hACE2 and PWK x K18-hACE2 at (**f**) 3 dpi and (**g**) 6 dpi. Correlation analyses were performed by computing Spearman's rank correlation coefficients. Correlation tests with $p < 0.001$ were displayed.

calling of cell types suggested that a significant proportion of Visium spots captured transcriptional signatures representing more than one cell type, predominantly endothelial cells and pneumocytes as expected in the lung (Fig. 4a, d). In K18-hACE2 mice, transcriptional changes following infection were relatively modest, with the largest change in gene signatures associated with myeloid cells, classical monocytes, and alveolar macrophages only by 6 dpi (Fig. 4b, c). In contrast, PWK x K18-hACE2 mice demonstrated dynamic changes in the lung, including a marked stromal cell response as well as recruitment of classical monocytes at 3 dpi that was resolving by 6 dpi (Fig. 4e). Evidence of T cell infiltration was also observed at 3 dpi in gene cluster 3 that contained monocytes and endothelial cells (Fig. 4d, f). In contrast, endothelial cell gene expression was reduced in representation at both 3 and 6 dpi compared to naïve lungs. However, this is likely due to increased cellular complexity of the lung at 3dpi, and a greater transcriptional signature of type II pneumocytes by 6 dpi (Fig. 4d) which may indicate cellular proliferation and repair at this time. These results demonstrate an early, orchestrated tissue-level response in PWK x K18-hACE2 mice resulting in coordinated monocyte and T cell infiltration, that was substantially delayed in the K18-hACE2 mouse model.

Previous studies in C57BL/6 mice have demonstrated a role for IFNAR signaling in immune cell recruitment to the lung, but little to no role in control of SARS-CoV-2 replication or protection from clinical disease[15,16]. From the results obtained thus far, we suspected that low levels of IFN-I expression in K18-hACE2 at early time points enables uncontrolled virus replication but also renders this strain difficult to study the antiviral functions of IFN-I. To explore this further, expression dynamics of ISGs previously demonstrated to have direct antiviral functions towards SARS-CoV-2[45] were examined. Like the monocyte response in PWK x K18-hACE2 mice, expression of representative ISGs *Ifi205, Zbp1, Ifit3, Ly6e, Gbp3, and Bst2* uniformly peaked at 3 dpi and was resolving by 6 dpi (Fig. 5a). ISG expression changes were generally apparent across all transcriptional clusters, suggesting that no single cell type was uniquely responsible. In contrast, ISG expression in K18-hACE2 mice was not uniformly orchestrated and was generally muted compared to PWK x K18-hACE2 mice. Low levels of *Ifi205, Zbp1, and Gbp3* expression at 3 dpi were further increasing at 6 dpi, whereas increased *Ifit3* and *Bst2* expression at 3 dpi was sustained at 6 dpi, and *Ly6e* expression was not changed over time. The dynamics of ISG expression appeared closely related to virus clearance. By RNAscope analysis for SARS-CoV-2 nucleic acid, the virus was similarly distributed throughout the lungs of both mouse genotypes at 3 dpi (Fig. 5b). By 6 dpi, virus distribution was unchanged in K18-hACE2 mice but was almost completely cleared from PWK x K18-hACE2 lungs. Thus, the comparison between these highly sensitive and resistant genetic backgrounds reveals that orchestrated IFN-I and monocyte responses are closely associated with control of virus replication.

The distinct contrasts between K18-hACE2 and PWK x K18-hACE2 mice in ISG signatures, chemokine expression, and viral dynamics

prompted us to test the role of IFN-I by treating both genetic backgrounds with a single dose of anti-IFNAR neutralizing monoclonal antibody 24 h prior to infection. For these studies, K18-hACE2 and PWK x K18-hACE2 were inoculated with a lethal dose ($10^4$ pfu of SARS-CoV-2), with male PWK x K18-hACE2 maintaining a highly resistant phenotype (Fig. S2a). Consistent with correlation coefficient analysis, neutralization of IFN-I signaling in male mice resulted in reduced expression of most inflammatory cytokines and chemokines in BAL fluid at 3 and 6 dpi regardless of genetic background (Fig. 6a). Exceptions in PWK x K18-hACE2 mice included increased IFNα and IL-12p70 at 3dpi, and no change in IFNγ and IL-18 at 6 dpi. Virus titers in lungs of PWK x K18-hACE2 mice were increased by approximately 8-fold at 3dpi, with virus clearance delayed in most mice, increased neuroinvasion, and increased lethality (Fig. 6b–d). In contrast, while neutralization of IFN-I signaling resulted in a 5-fold increase in infectious titers in C57Bl/6 K18-hACE2 lung at 3dpi, it did not change the kinetics of virus clearance or rates of survival (Fig. 6d). Thus, early IFN-I expression in PWK x K18-hACE2 controls peak virus loads, kinetics of virus clearance and dissemination to tissues outside of the lung. Importantly, this work definitively demonstrates the antiviral role of IFN-I to SARS-CoV-2 in vivo and illustrates the role of host genetics in determining these critical events.

## Discussion
Although the COVID-19 pandemic has spurred unprecedented analyses of human cohorts, the drivers of dysregulated inflammation and how they relate to virus replication kinetics in tissues are not understood. Thus, animal models that recapitulate the complexities of human COVID-19 virus replication dynamics and inflammatory responses are essential to better understand mechanisms of protection and pathogenesis. Here, we report an expanded range of SARS-CoV-2 disease phenotypes in mice using F1 progeny from the eight CC founder strains, as well as BALB/cJ and DBA/2 J, crossed to K18-hACE-2 transgenic mice. The various genetic backgrounds yielded a spectrum of disease phenotypes including severe (C57BL/6 J and A/J), severe with sex-bias (CAST/EiJ, NOD/ShiLtJ and WSB/EiJ), and resistant mice exhibiting relatively high pathology but non-lethal disease (PWK/PhJ, NZO/HILtJ, 129S1/SvImJ, DBA/2 J and BALB/cJ). Variability in outcomes of SARS-CoV-2 infection in these mouse models was associated with distinct dynamics of innate immunity. Herein, control of virus replication in the lung and host resistance to disease can be independent of the early peak titer, but it was closely associated with a phased amplification and resolution of pro-inflammatory mediators associated with rapid control of virus replication in lung and prevention of virus dissemination to other organs. These favorable outcomes of COVID-19 are most clearly modeled by examining SARS-CoV-2 infection of PWK and NZO x K18-hACE2. In contrast, infection of C57BL/6 or A/J x K18-hACE2 represents a model of inefficient IFN expression in the lung, failed host control of virus replication, and dysregulated proinflammatory responses[8,10,43,46–48]. Collectively, the CC x K18-hACE-2

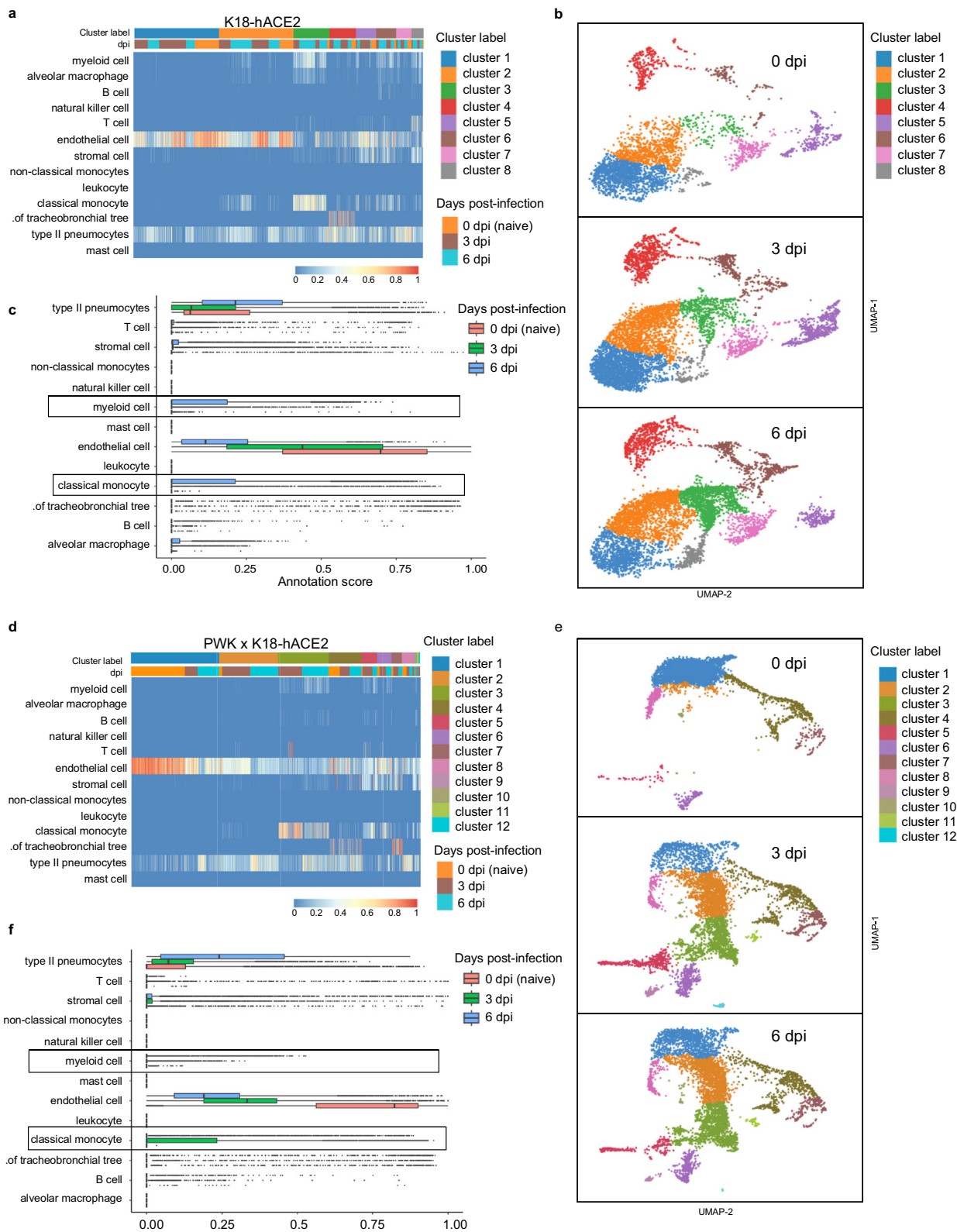

**Fig. 4 | Spatial transcriptomics reveals differential innate immune responses of pneumocytes, endothelial cells and monocytes in K18-hACE2 and PWK x K18-hACE2 mice.** 10x Genomics Visium spatial transcriptomic analysis of lung sections from K18-hACE2 and PWK x K18-hACE2 at 0 (naïve), 3 and 6 dpi following SARS-CoV-2 inoculation. **a**, **d** Confidence scores for cell type annotations based on unsupervised cluster analysis in K18-hACE2 (**a**) and PWK x K18-hACE2 (**d**). **b**, **e** Bar graph of cell annotations scores in K18-hACE2 (**b**) and PWK x K18-hACE2 (**e**). **c**, **f** UMAP representation of identified transcriptional clusters in K18-hACE2 (**c**) and PWK x K18-hACE2 (**f**).

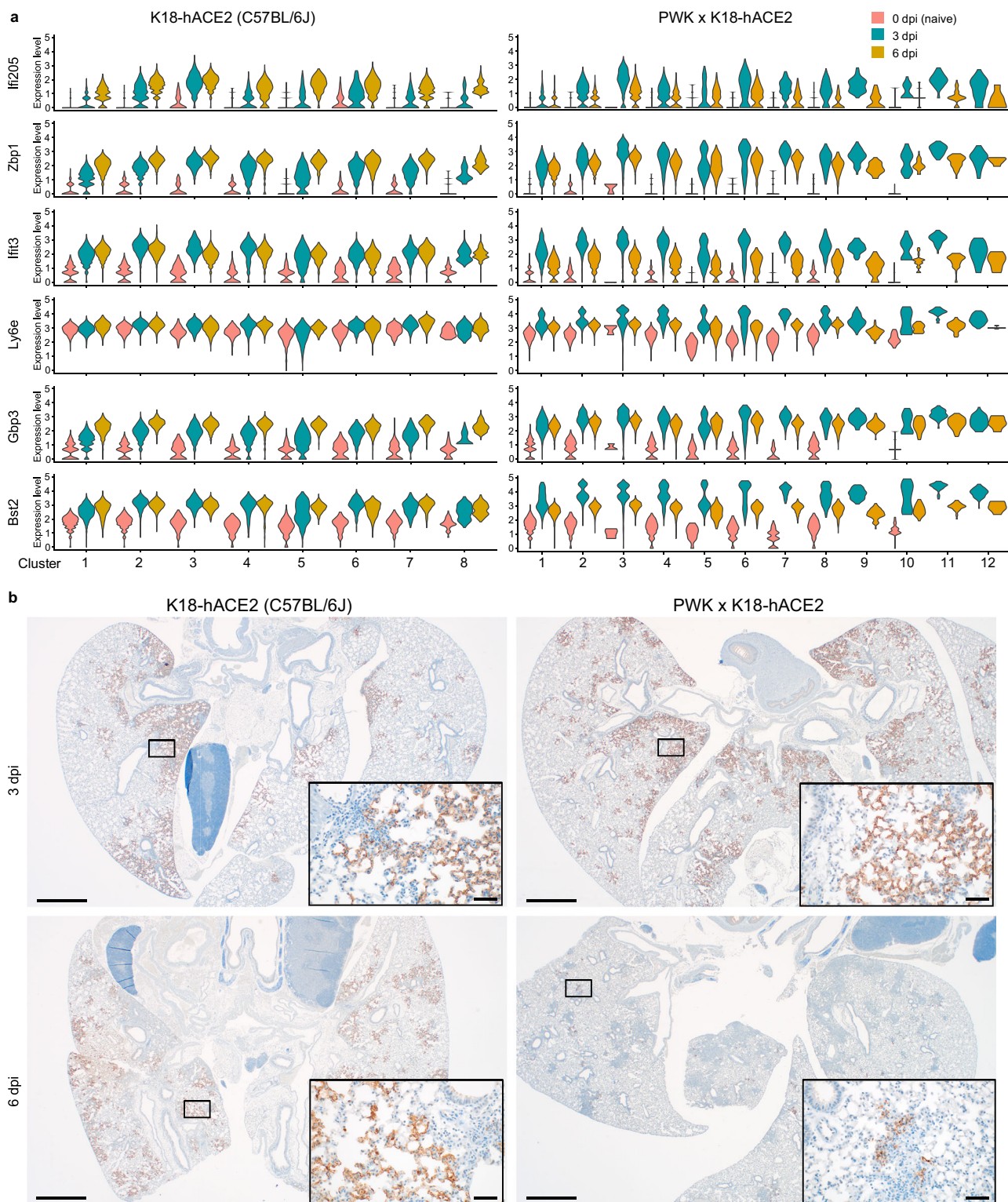

**Fig. 5 | Rapid kinetics of ISG expression associated with virus clearance in PWK x K18-hACE2. a** Violin plots depicting expression levels of individual ISGs identified in Visium spatial transcriptomics analysis. **b** In-situ hybridization for SARS-CoV-2 nucleic acid in lung from male K18-hACE2 and PWK x K18-hACE2 mice at 3 and 6 dpi.

Scale bars represent 1000 μm; scale bars in insets represent 50 μm. Biological replicates were examined over at least two experiments and 3 male mice/strain/timepoint were randomly selected for assessment of virus distribution by in-situ hybridization. Images are representative of biological replicates.

panel represents a genetically diverse population and reflects a broader range of host-viral interactions that influence disease outcomes. Importantly, this panel provides much-needed tools to further determine the roles of specific cell types and signaling pathways in orchestrating protective versus pathogenic immunity.

Early events surrounding the production of antiviral IFNs are key determinants in human outcomes of COVID-19, but a mechanistic understanding is lacking. Evidence linking IFN-I responses to dysregulated immunity and disease severity includes genetic lesions in pattern recognition receptor signaling molecules[3], production of anti-IFN

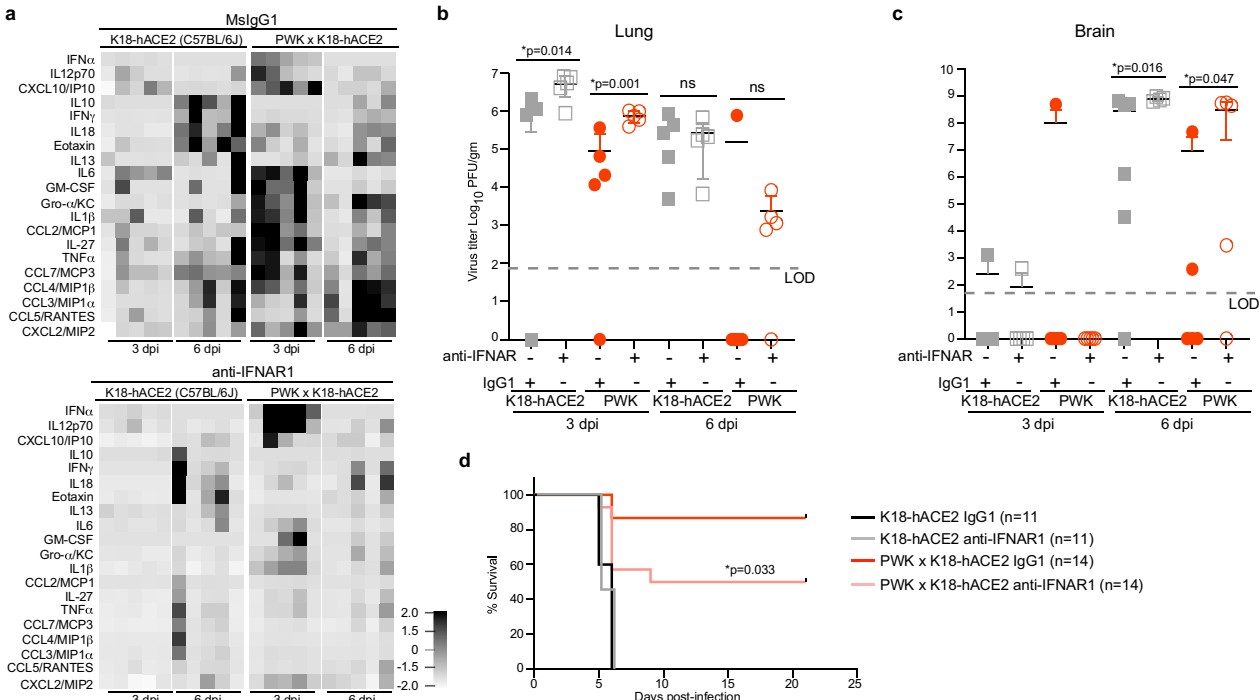

**Fig. 6 | Type I IFN signaling controls early SARS-CoV-2 replication and coordinates inflammatory responses in PWK x K18-hACE2 mice.** Male K18-hACE2 and PWK x K18-hACE2 mice were treated with anti-IFNAR or msIgG1 isotype control 1 day prior to intranasal inoculation with $10^4$ pfu of SARS-CoV-2. **a** Heatmap of cytokine and chemokine levels in BAL collected at 3 and 6 dpi. **b, c** Virus titers (pfu/g tissue) in lung (**b**) and brain (**c**) at 3 and 6 dpi. Graphs represent virus titers from individual mice (n = 5 mice/timepoint/experimental group) with the mean ± SD.

Two-tailed, unpaired Student's *t*-test was used to compare virus titers. *$p < 0.05$ was considered statistically significant. **d** Percent survival of mice followed daily until 21 dpi. Biological replicates were examined over two independent experiments, the total number of mice per group was as follows: (K18-hACE2, IgG1 n = 11; K18-hACE2, anti-IFNAR1 n = 11; PWK x K18-hACE2, IgG1 n = 14; PWK x K18-hACE2, anti-IFNAR1 n = 14). The Mantel-Cox log-rank test with 95% confidence interval was used to compare survival curves for IgG1- versus anti-IFNAR1-treated groups.

autoantibodies[12], and patterns of IFN-I and ISG expression in respiratory tissues[49]. However, these attributes are linked to subsets of patients with severe COVID-19, which leaves open questions regarding how antiviral functions of IFNs are balanced with inflammation and why some people fail to mount effective antiviral responses despite high viral load or high IFN-I expression. The CC x K18-hACE2 mice revealed that IFN-I responses in lungs of K18-hACE2 (C57BL/6 J) are relatively low compared to other specific genetic backgrounds, which may explain why ablation of IFN-I signaling was associated with relatively small effects in this model (this study, and[15,16]). In contrast, the PWK x K18-hACE2 infection model represents the outcome where IFN-I is directly linked to control of peak virus loads, the magnitude of inflammatory cytokine expression independent of virus burden, regulation of adaptive immunity, virus clearance, and control of virus dissemination to tissues outside of the lung. It is curious then that IFN-I expression was highest in BAL fluid of WSB x K18-hACE2 that had high early virus burden in the lung with delayed clearance similar to mice with low IFN-I expression (C57BL/6J-K18-hACE2). Thus, by comparison, viral dynamics, and pathology in WSB x K18-hACE2 mice provide a model of high IFN-I driving pathology combined with inefficient antiviral functions, as these mice had some of the highest lung pathology scores. Interestingly, a survey of immune cell populations in CC founder mice demonstrated that unchallenged WSB mice have the highest percentages of plasmacytoid dendritic cells (pDCs)[50,51], critical producers of IFN-I in blood and tissues. It is possible that the cellular origin of IFN-I in lungs of SARS-CoV-2-infected PWK and WSB x K18-hACE2 mice is different, that pDC function in the WSB background is dysregulated as is observed in severe COVID-19[52], or that high IFN-I expression and resulting pathology or inflammatory cell milieu inhibits adaptive immunity in WSB mice. Another consideration is that SARS-CoV-2 may differentially influence IFN production and signaling in various genetic backgrounds. Further

insight will be gained by comprehensively comparing immunological and transcriptional responses of PWK and WSB x K18-hACE2 mice following the challenge with SARS-CoV-2. Importantly, virus dissemination to the CNS does not consistently occur in PWK or WSB genetic backgrounds suggesting that these two new models are particularly valuable in understanding immune cell dynamics in the lung and should be explored for their potential to model post-acute events including sequalae of long-COVID.

The remarkably different patterns of ISG expression in the lungs of K18-hACE2 and PWK x K18-hACE2 mice begin to clarify how IFN-I signaling can fail to be well orchestrated and effective. We examined canonical ISGs with demonstrated antiviral functions towards SARS-CoV-2 (e.g., Bst2/tetherin and ZBP1[45]), as well as Gbp proteins that SARS-CoV-2 VOC have specifically evolved resistance to[53]. Prompt and widespread gene expression was tightly associated with efficient control of virus replication in PWK x K18-hACE2 lungs. However, while ISG expression was increased at 3 dpi of K18-hACE2 mice, it failed to effectively control virus replication resulting in further increases in host gene expression. We speculate that this sustained ISG expression in the presence of a relatively high virus burden may provide an environment that facilitates virus evolution to escape individual ISGs. As expected, IFN-I was not solely required for control of virus replication in PWK x K18-hACE2 mice as IFNAR blocking experiments did not result in a rebound of virus titers to the same levels as K18-hACE2 or 100% lethality. However, the PWK genetic background will enable experimental models to empirically determine how antiviral mechanisms are coordinated to control replication, including type III IFNs that have been shown to correlate with virus control in humans[49].

One limitation of the study is the degree of variability observed in the data likely due to the sublethal virus dose used. However, as exemplified in PWK x K18-hACE2, this was an important element of the

experimental design because it enabled observation of host resistance by not overwhelming initial host defenses. Another limitation is the use of heterozygous F1 offspring because important phenotypes could be due to either or both parental alleles, making it difficult to map the genetic loci responsible for a given phenotype. However, K18-hACE2 mice were used as a common parent for all F1s and therefore, any differences observed in disease outcome are due to genes contributed by the unique parental strain (CC founders, BALB/cJ, or DBA/2).

In summary, the phenotypes in diverse mice support a role for the innate immune system in determining COVID-19 severity[3,8,46–48] and can be used to address key knowledge gaps, including mechanisms of innate immune control of virus replication, defining events needed for a well-orchestrated inflammatory response independent of early viral burden, molecular mechanisms of sex-dependent disease severity, and longer-term implications for tissue repair[54] and lung function. Taken together, these observations demonstrate that the use of host genetic variation in mice can model different outcomes of SARS-CoV-2 infection, addressing a major deficiency in the toolkit required to combat COVID-19.

## Methods

### Ethics statement

Animal study protocols were reviewed and approved by the Institutional Animal Care and Use Committee (IACUC) at Rocky Mountain Laboratories (RML), NIAID, NIH in accordance with the recommendations in the Guide for the Care and Use of Laboratory Animals of the NIH. All animal experiments were performed in an animal biosafety level 3 (ABSL3) research facility at RML. Standard operating procedures for work with infectious SARS-CoV-2 and protocols for virus inactivation were approved by the Institutional Biosafety Committee (IBC) and performed under BSL3 conditions.

### Virus preparation

SARS-CoV-2 (USA_WA1/2020) from the University of Texas Medical Branch (Vero passage 4) was propagated on Vero cells cultured in DMEM supplemented with 10% fetal bovine serum, 1 mM L-glutamine, 50U/ml penicillin and 50ug/ml streptomycin. Culture supernatants were collected at 72 hpi, aliquoted, and stored at −80 °C.

### Mice, virus infection, and in vivo block of IFNAR1

CC founder x C57BL/6 J -K18-hACE2 F1 were provided by The Jackson Laboratories and include the following: B6.Cg-Tg (K18-ACE2)2PrImn/J), 034860; (A/J x B6.Cg-Tg(K18-ACE2)2PrImn/J)F1/J, 035940; (PWK/PhJ x B6.Cg-Tg(K18-ACE2)2PrImn/J)F1/J, 035938; (NZO/HlLtJ x B6.Cg-Tg(K18-ACE2)2PrImn/J)F1/J, 035936; (129S1/SvImJ x B6.Cg-Tg(K18-ACE2)2PrImn/J)F1/J, 035934; (CAST/EiJ x B6.Cg-Tg(K18-ACE2)2PrImn/J)F1/J, 035937; (NOD/ShiLtJ x B6.Cg-Tg(K18-ACE2)2PrImn/J)F1/J, 035935; (WSB/EiJ x B6.Cg-Tg(K18-ACE2)2PrImn/J)F1/J, 035939; (BALB/cJ x B6.Cg-Tg(K18-ACE2)2PrImn/J)F1/J, 035941; (DBA/2 J x B6.Cg-Tg(K18-ACE2)2PrImn/J)F1/J 035943. Six- to 12-week-old male and female mice were inoculated by the intranasal route with $10^3$ pfu of SARS-CoV-2 in a volume of 50ul PBS (Gibco). Prior to inoculation mice were anesthetized by inhalation of isoflurane. Mice were monitored daily for clinical signs of disease and weight loss. In IFNAR1 blocking experiments, male K18-hACE2 and PWK x K18-hACE2 mice were inoculated intraperitoneally with anti-IFNAR1 (2.0 mg/mouse, MAR1-5A3, Bio X Cell, Inc.) or msIgG1 isotype control (2.0 mg/mouse, MOPC-21, Bio X Cell, Inc.) 1 day prior to intranasal inoculation with $10^4$ pfu of SARS-CoV-2.

Prior to and during experiments, same-sex, group-housed animals were kept at 12-hour light cycle, $22 \pm 2$ °C, and 40–60 % humidity in autoclaved individually-ventilated cages (Super Mouse 750™ AllerZone™ Micro-Isolator® Ventilated Rack, Lab Products LLC. Aberdeen, MD) with autoclaved bedding (Sani-Chips®, P.J. Murphy Forest Product Corp., Montville, NJ), and provided with *ab libitum* rodent chow (2016 Teklad Global 16% protein rodent diet, Envigo Teklad, Denver, CO) and reverse osmosis water. Enrichment included nesting material (Nestlets by Ancare, Ancare Corp. Bellmore, NY) and additional shelters as needed (Shepherd Shack®, Shepherd Specialty Papers, Watertown, TN).

### Virus titration

Infectious virus in lung and brain tissue was quantified by plaque assay. Tissues were collected in 0.5 ml of DMEM containing 2% FBS, 50U/ml penicillin, and 50ug/ml streptomycin and immediately frozen. Tissue samples were weighed, and then homogenized using 5 mm steel beads and TissueLyzer II high-speed shaker (Qiagen). Ten-fold serial dilutions of homogenates were prepared in duplicate and used to inoculate Vero cells grown in 48-well tissue culture plates. Following 1 h of incubation at 37 °C, the cells were overlayed with 1.5% carboxymethyl cellulose (CMC) in MEM and incubated at 37 °C for 3-4 days. Cells were then fixed in 10% formalin and plaques were visualized by staining with 1% crystal violet diluted in 10% ethanol.

### Measurement of SARS-CoV-2-specific IgG

Sera were collected at 21 dpi from mice that survived SARS-CoV-2 infection. SARS-CoV-2 spike protein-specific IgG was measured using SARS-CoV2 spike protein serological IgG ELISA kit (Cell Signaling Technology) per manufacturer's instructions.

### Multiplex cytokine/chemokine analysis

Sera were collected from SARS-CoV-2-infected mice at 3 and 6 dpi by centrifugation of whole blood in GelZ serum separation tubes (Sarstedt). BAL samples were recovered by insufflation of lungs with 1 ml sterile PBS followed by aspiration to collect ~0.5 ml volume of fluid. SARS-CoV-2 in sera and BAL fluid were inactivated by using γ- irradiation (2 MRad) and removed from the BSL3 laboratory. Cytokine concentrations in serum and BAL were measured using a 26-Procartaplex mouse cytokine/chemokine panel (ThermoFisher EPX260-26088-901) combined with a Simplex IFN*a* assay (ThermoFisher EPX01A-26027-901). Samples were run on a Luminex Bio-Plex 200 system with BioPlex Manager 6.1.1 software. Data represented are from samples collected from individual mice.

### Histopathology and in situ hybridization

Tissues were fixed in 10% neutral buffered formalin for a minimum of 7 days with 2 changes according to IBC-approved standard operating procedure. Tissues were processed with a Sakura VIP-6 Tissue Tek, on a 12-hour automated schedule, using a graded series of ethanol, xylene, and PureAffin. Embedded tissues were sectioned at 5 µm and dried overnight at 42 °C prior to staining with hematoxylin and eosin. Chromogenic detection of SARS-CoV-2 viral RNA was performed using the RNAscope VS Universal AP assay (Advanced Cell Diagnostics Inc.) on the Ventana Discovery ULTRA stainer using a SARS-CoV-2 specific probe (Advanced Cell Diagnostics Inc. cat. 848569). In situ hybridization was performed according to the manufacturer's instructions. ACE2 immunoreactivity was detected using R&D Systems AF933 at a 1:100 dilution. The secondary antibody is the Vector Laboratories catalog number MP-7405 ImmPress anti-goat polymer. The tissues were then processed for immunohistochemistry using the Discovery Ultra automated stainer (Ventana Medical Systems) with a Chromo-Map DAB kit (Roche Tissue Diagnostics cat. 760–159).

### Visium spatial transcriptomics and processing

Formalin-fixed, paraffin-embedded samples were sectioned directly onto Visium Spatial for FFPE Gene Expression Mouse Transcriptome slides (6.5 ×6.5 mm) and processed for sequencing, according to the manufacturer's procedure (10X Genomics, Pleasanton, CA). The resulting libraries were sequenced as 28/10/10/50 base reads on the NovaSeq instrument using the S2 Reagent kit in paired-end mode (Illumina, San Diego, CA). Fastq files were processed with the Spaceranger-1.4.2 pipeline. Processed data were then analyzed on R

studio v4.1.3 (2021.09.1, Build 372) with Seurat (v4.0)[55]. We trimmed spots with zero counts and normalized using SCtransform function[56]. Then data were integrated using IntegrateAnchor method[57]. A total of 2000 highly variable genes identified by the SCtransform function in the Seurat[55] package were used for principal component analysis (PCA)-based dimensionality reduction with RunPCA. Uniform Manifold Approximation and Projection (UMAP) with a resolution of 0.05 was utilized to visualize single-cell clustering using principal components (PCs) 1 to 30.

## Quantitative real-time PCR

Total lung RNA was extracted from homogenized lysates using the *mir*Vana miRNA Isolation Kit (Invitrogen cat. AM1560) following the manufacturer's protocol for total RNA isolation. Total RNA samples were quantified using a Nanodrop spectrophotometer (Thermo cat. ND-2000C), and reverse transcribed to cDNA using the SuperScript IV VILO Master Mix with ezDNase enzyme (Invitrogen cat. 11766050) following the manufacturer's instructions. Expression of mouse *Ace2*, mouse *Gapdh*, and human ACE2 were quantified by qPCR using the ViiA7 Real-Time PCR System (Thermo cat. 44535545). TaqMan probes targeting mouse *Gapdh* (Thermo probe ID Mm99999915_g1) and human ACE2 (Thermo probe ID Hs01085333_m1) were used in conjunction with the TaqMan Fast Advanced Master Mix (Thermo cat. 4444556) and the ViiA7 Real-Time PCR System with QuantStudio Software v1.7.2 (Thermo cat. 44535545). Samples were run in technical triplicate in 10ul reaction volumes on a 384-well plate, and hACE2 transcript abundance values were normalized to m*Gapdh* in each sample and relative expression was calculated using a reference lung sample (K18-hACE2) and the delta delta-Ct method.

## Statistics & reproducibility

Comparison of survival curves in males versus females for each strain was performed using the log-rank (Mantel-Cox) test. Two-tailed student's t-test Holm-Bonferroni multiple comparison posttest was used to compare cytokine and chemokine levels of BAL and serum. Kruskal-Wallis test with Dunn's posttest was used to compare lung virus titers in each strain/sex to that of K18-hACE2. Differences between groups were considered significant at a *p*-value of <0.05. All statistical analyses were performed with graphPad Prism 8.0 (GraphPad Software) or Qlucore Omics Explorer Version 3.7 (Qlucore AB). Correlation analyses were performed by computing Spearman's coefficients and visualized using ggcorrplot. Correlation tests with $P < 0.001$ were displayed.

## Reporting summary

Further information on research design is available in the Nature Portfolio Reporting Summary linked to this article.

# Data availability

All data are available without restrictions. Visium RNA-seq data that support the findings of this study have been deposited in Gene Expression Omnibus (GEO) public database with the accession code GSE231711. Source data are provided in the Source Data file. Source data are provided with this paper.

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

## Acknowledgements
This work was funded by the Division of Intramural Research, National Institutes of Health, National Institute of Allergy and Infectious Diseases. Thank you to the animal caretaker staff at campuses of both RML and The Jackson Laboratory for their work in animal husbandry.

## Author contributions
Conceptualization: S.B., N.R., and S.M.H., Methodology: S.J.R., N.R., S.B., and C.M., Investigation: S.J.R., O.B., K.L.M., C.S., C.C., M.L., R.M.B., A.I.C., J.G.S., G.L.S., R.R., S.L.A., E.F., C.P., C.N.B., C.B., S.M., D.P.B., J.B.L., J.M.L., A.S., P.G., D.E.S., and J.S., Visualization: S.J.R., C.N.B., J.H., D.E.S., C.M., D.P.B., J.B.L., J.M.L., A.S., P.G., and J.S., Funding acquisition: S.B. and N.R., Supervision: S.B. and N.R., Writing – original draft: S.B., S.J.R., C.B., A.S., and N.R., Writing – review & editing: S.B., S.J.R., C.B., and N.R.

## Funding

## Competing interests
The authors declare no competing interests.
