## [Peer Review File · Nature Communications]

REVIEWER COMMENTS

Reviewer #1 (Remarks to the Author):

Animal models are an indispensable tool to understand infection dynamics and model disease phenotypes during virus infection. In this study, the authors crossed the Collaborative Cross founder strains to K18-hACE2 transgenic mice, thereby generating a genetically diverse mouse panel that has been previously shown to replicate the interpersonal genetic variance in humans. This study is clearly written and presents compelling evidence regarding the usefulness of this system to understand the complicated processes governing COVID-19. However, there are several limitations, most importantly the lack of immune cell analyses. A previous study has already demonstrated the usefulness of the Collaborative Cross Mice for coronavirus research (<https://doi.org/10.1371/journal.ppat.1009287>), even though the authors were using SARS-CoV, this study allowed to determine immune signatures of severe COVID-19 disease and provided supporting evidence for the role of T cells. More information about infiltrating immune cells into the lungs should be provided. Similarly, the study would strengthen significantly upon inclusion of specific immune cell analyses via FACS. While this study descriptively and very comprehensively analyzed the cytokine profiles caused by viral infection, there are no mechanistic data included, such as analyzing the genetic loci possibly contributing to the observed phenotypes.

Minor points: In accordance with the journal guidelines the authors should avoid "data not shown" statements and instead include data necessary to evaluate the claims of the paper as Supplementary Information. Sup. Fig.S4: Please include the bar for the expression levels (-2 to 2).

Reviewer #2 (Remarks to the Author):

The study entitled 'Genetically diverse mouse models of SARS-CoV-2 infection reproduce clinical 1 variation and cytokine responses in COVID-19' by Robertson et al. describes a mouse panel with different genetic diversity and clinical outcomes (susceptibility and resistance) to SARS-CoV-2 infection. This is a comprehensive study, looking at morbidity, mortality, viral titers and immunomodulatory responses to the infection studied. Some correlations are described between mouse strains that are resistant or susceptible to SARS-CoV-2 infection verifying the dynamics of this infection in humans.

A major concern is the lack of power in some of the observed results (e.g. PFU values in several tissues, pneumonia score), mainly because in each mouse strain studied there are mice that behaved differently, and this complicates the interpretation of the results. This complexity is understandable and can be discussed as a limitation of the models, so future research can account for it. This could be also linked to the low viral infection dose used (10^3) with respect to other published models.

If the full story goes around the levels of IFN- α , which at higher levels produced in the lungs provides resistance to the infection, a validation of the role of IFN- α study could be performed on mouse resistant mice.

Authors chose to use F1 progeny and this has its own limitations that can be introduced and discussed.

The relative expression of human ACE2 in the lungs of naive CCxK18-hACE2 F1 progeny is presented. However, it is not clear if the distribution within the tissue is equal in each studied mouse strain and if this tissue distribution correlates with the severity of the infection observed for each mouse model.

Authors may consider moving Fig. S4 to the main body.

Unsupervised clustering results presented in Figs 3D and E are interesting but difficult to interpret because infections did not show homogeneous results within each mouse strain studied. For example, resistant mice in cluster 2 represent 52.6% of the total mice in this cluster, and in cluster 3 resistant mice represent 44.7% of the mice in this cluster. Thus, the phenotype that defines these two clusters (low and high production of cytokines) is not so obvious that depends on the resistant mouse phenotype. As there are mice within the same mouse strain that are resistant or sensitive (susceptible) to the infection, it is not clear if the resistant and sensitive mice within each mouse strain studied is due to these being differentially infected at the time of intranasal infection.

Revise some of the missing labels in graphs, especially in supplemental figures.

Revise text, some sentences are unfinished (line 321)

Reviewer #1 (Remarks to the Author):

Animal models are an indispensable tool to understand infection dynamics and model disease phenotypes during virus infection. In this study, the authors crossed the Collaborative Cross founder strains to K18-hACE2 transgenic mice, thereby generating a genetically diverse mouse panel that has been previously shown to replicate the interpersonal genetic variance in humans. This study is clearly written and presents compelling evidence regarding the usefulness of this system to understand the complicated processes governing COVID-19. However, there are several limitations, most importantly the lack of immune cell analyses. A previous study has already demonstrated the usefulness of the Collaborative Cross Mice for coronavirus research (<https://doi.org/10.1371/journal.ppat.1009287>), even though the authors were using SARS-CoV, this study allowed to determine immune signatures of severe COVID-19 disease and provided supporting evidence for the role of T cells. More information about infiltrating immune cells into the lungs should be provided. Similarly, the study would strengthen significantly upon inclusion of specific immune cell analyses via FACS. While this study descriptively and very comprehensively analyzed the cytokine profiles caused by viral infection, there are no mechanistic data included, such as analyzing the genetic loci possibly contributing to the observed phenotypes.

Reply: We appreciate the Reviewer's suggestions and agree that greater characterization of these models is needed. As this is the first report of these CC x K18-hACE2 models, we elected to use 10XGenomics Visium spatial transcriptomics platform to obtain an unbiased, global assessment of changes not only in immune cell populations, but also in resident lung cells such as endothelial cells, type I and type II pneumocytes, and stromal cells that play important roles SARS-CoV-2 infection and response. For this reason, this approach offered an advantage over standard flow cytometry focused on defining infiltrating immune cells. This approach also enabled us to identify the dynamics of IFN-stimulated gene expression across lung cellular populations and to provide mechanistic data on the role of the IFN-I response in control of virus replication and inflammation. To this end, we focused on K18-hACE2 (sensitive) and PWK x K18-hACE2 (resistant) mice at 0, 3 and 6 dpi to gain better insight into the kinetics of responses associated with different outcomes of infection. Lung samples from male mice were used because the PWK x K18-hACE2 males showed a greater increase in IFN-I in the BAL compared to females (Fig. 3a) and they also maintained a resistant phenotype with a higher virus dose (10^4 pfu/mouse) (Fig. S2a). As shown in Fig. 4, peak classical monocyte infiltration the lung occurred at 3 dpi in PWK x K18-hACE2 and was nearly resolved by 6 dpi. In contrast, monocyte responses in K18-hACE2 mice were delayed, but the inflammatory response was more sustained as indicated by greater numbers of classical monocytes, alveolar macrophages and other myeloid cells present in lung at 6 dpi. These kinetic differences were consistent with the cytokine and chemokine responses observed in BAL (Fig. 3d). Importantly, this new analysis has yielded tissue-level information on coordination of early immune responses to SARS-CoV-2 infection, and addressed critical questions in the field by demonstrating how kinetics of the IFN-I response is associated with virological control and inflammation.

To address the mechanistic aspect of these phenotypes, we feel that genetic mapping is a long-term goal not feasible in this timeframe, but we did evaluate the role of IFN-I in K18-hACE2 (sensitive) and PWK x K18-hACE2 (resistant) mice by performing IFN-I blocking experiments. K18-hACE2 and PWK x K18-hACE2 male mice were treated with anti-IFNAR blocking antibody or isotype control 24 hours prior to SARS-CoV-2 inoculation. In both strains, a block in IFN-I signaling resulted in reduced expression of most cytokines and chemokines in BAL at 3 and 6 dpi (Fig. 6b). Anti-IFNAR treated PWK x K18-hACE2 had significantly higher virus titers in the lung at 3 dpi, delayed virus clearance at 6 dpi, increased virus titers in the brain and increased lethality (Fig. 6c-e). These data demonstrate that type I IFN signaling controls SARS-CoV-2 replication and tissue dissemination, as well as coordinates inflammatory responses in PWK x K18-hACE2 mice. Importantly, this work also demonstrates the lack of an effective early IFN-I response in K18-hACE2 and explains why clear antiviral functions of IFN-I have been difficult to demonstrate in the K18-hACE2 model.

Minor points: In accordance with the journal guidelines the authors should avoid "data not shown" statements and instead include data necessary to evaluate the claims of the paper as Supplementary Information. Sup. Fig.S4: Please include the bar for the expression levels (-2 to 2).

Reply: Minor points:

- 1) We have removed “data not shown” and replaced it with the statement:
“Infection was confirmed in all survivors by seroconversion to SARS-CoV-2 nucleoprotein.” Line 101
- 2) Reference bar for heatmap has been added to Fig. S4.

Reviewer #2 (Remarks to the Author):

The study entitled ‘Genetically diverse mouse models of SARS-CoV-2 infection reproduce clinical 1 variation and cytokine responses in COVID-19’ by Robertson et al. describes a mouse panel with different genetic diversity and clinical outcomes (susceptibility and resistance) to SARS-CoV-2 infection. This is a comprehensive study, looking at morbidity, mortality, viral titers and immunomodulatory responses to the infection studied. Some correlations are described between mouse strains that are resistant or susceptible to SARS-CoV-2 infection verifying the dynamics of this infection in humans.

A major concerns is the lack of power in some of the observed results (e.g. PFU values in several tissues, pneumonia score), mainly because in each mouse strain studied there are mice that behaved differently, and this complicates the interpretation of the results. This complexity is understandable and can be discussed as a limitation of the models, so future research can account for it. This could be also linked to the low viral infection dose used ($10E3$) with respect other published models.

Reply: We agree with the reviewer’s comment regarding the variability of the data and its sources. In the initial studies, we chose the sublethal dose in K18-hACE2 precisely to enable a wide range of outcomes, and we believe that this is also a strength of the study as lethal viral doses may overcome host resistance particularly as it relates to early innate immunity. As requested, we have included this as a discussion point regarding the limitations of the study (Discussion, lines 338-344). We have also begun to address this experimentally by comparing the survival in K18-hACE2 and PWK x K18-hACE2 using a higher virus dose (10^4 pfu/mouse). As shown in Fig. S2a, all K18-hACE2 reached endpoint criteria by 5-7 dpi demonstrating that this dose was uniformly lethal in a sensitive strain. Although female PWK x K18-hACE2 showed greater sensitivity to the higher dose compared to 10^3 (60% lethality at 10^4 pfu compared to 20% at 10^3 pfu (Fig. 1b)), male PWK x K18-hACE2 maintained a resistant phenotype with 90-100% surviving infection (Fig. S2a). We then used this more stringent approach in male mice to demonstrate the importance of the innate IFN-I response in controlling downstream inflammation and viral control in PWK x K18-hACE2 mice. This work attests to the biological relevance of our initial findings and further strengthens the manuscript.

If the full story goes around the levels of IFN-alpha, which at higher levels produced in the lungs provides resistance to the infection, a validation of the role of IFN-alpha study could be performed on mouse resistant mice.

Reply: We have addressed this central question in two ways. First, we performed 10XGenomics Visium spatial transcriptomics analysis on the most sensitive K18-hACE2 and resistant PWK x K18-hACE2 mice. This revealed the dynamics of IFN-stimulated gene expression and demonstrated that orchestrated IFN-I signaling across resident and infiltrating cells, as well as distinct monocyte dynamics, are closely associated with control of virus replication in PWK x K18-hACE2 mice. Second, we performed IFN-I blocking experiments in male K18-hACE2 and PWK x K18-hACE2 mice. We focused on male PWK x K18-hACE2 because they produced higher levels of IFN α than females (Fig. 3a) and they also maintained a strong resistant phenotype with a higher virus dose (10^4 pfu/mouse) (Fig. S2a). K18-hACE2 and PWK x K18-hACE2 male mice were treated with anti-IFNAR blocking antibody or isotype control 24 hours prior to SARS-CoV-2 inoculation. In both strains, a block in IFN-I signaling resulted in reduced expression of most cytokines and chemokines in BAL at 3 and 6 dpi (Fig. 6a). Anti-IFNAR treated PWK x K18-hACE2 had significantly higher virus titers in the lung at 3 dpi, delayed virus clearance at 6 dpi, increased virus titers in the brain and increased lethality (Fig. 6b-d). These data demonstrate that type I IFN signaling controls early SARS-CoV-2 replication and coordinates inflammatory responses in PWK x K18-hACE2 mice. Importantly, this work also

demonstrates the lack of an effective early IFN-I response in K18-hACE2 and explains why clear antiviral functions of IFN-I have been difficult to demonstrate in the K18-hACE2 model.

The relative expression of human ACE2 in the lungs of naive CCxK18-hACE2 F1 progeny is presented. However, it is not clear if the distribution within the tissue is equal in each studied mouse strains and of this tissue distribution correlates with the severity of the infection observed for each mouse model.

Reply: We appreciate this comment. To our knowledge, a species-specific antibody that detects human, but not mouse ACE2 is not commercially available. Therefore, to address the Reviewer's comment regarding the distribution of huACE2 in the CC x K18-hACE2 mice, we designed an RNAscope probe that would specifically recognize human ACE2 and not mouse ACE2 mRNA by *in situ* hybridization assay of formalin-fixed paraffin-embedded lung tissue from naive CC x K18-hACE2 mice. Unfortunately, we were unable to detect huACE2 RNA by *in situ* hybridization despite our exhaustive attempts and optimizations. Our only feasible option, albeit less ideal, was to use an ACE2 antibody that detects human and mouse ACE2 in immunohistochemical staining of lung tissue in a direct comparison between sensitive (K18-hACE2) and resistant (PWK x K18-hACE2) mice. As shown in Fig.S2c, both strains showed similar ACE2 staining predominantly in cells lining large bronchioles and low ACE2 staining in alveoli. Although this does not entirely rule the possibility that the hACE2 transgene expression is distributed differently in the two strains, similar transcriptional levels (Fig. S2b) and distribution of virus infection particularly at 3 dpi (Fig. 5b) suggests that the hACE2 expression is comparable in K18-hACE2 and PWK x K18-hACE2.

Unsupervised clustering results presented in Figs. 3D and E are interesting but difficult to interpret because infections did not show homogeneous results within each mouse strain studied. For example, resistant mice in cluster 2 represent 52.6% of the total mice in this cluster, and in cluster 3 resistant mice represent 44.7% of the mice in this cluster. Thus, the phenotype that define these two clusters (low and high production of cytokines) is not so obvious that depend on the resistant mouse phenotype. As there are mice within the same mouse strain that are resistant or sensitive (susceptible) to the infection, it is not clear if the resistant and sensitive mice within each mouse strain studied is due to these being differentially infected at the time of intranasal infection.

Reply: We agree that the inherent variability observed is likely due the low inoculating dose and intranasal inoculation route. The cytokine and chemokine assays were done on samples collected from the same mice shown in Fig. 2a, which indicates 1-2 animals per strain showing low infection, particularly at 3 dpi. We felt it was important to be transparent with all datasets as the variability likely stems from the experimental design. However, in the IFNAR blocking experiments, we have achieved a more direct demonstration of the biological importance of IFN-I in the induction of proinflammatory cytokines and relevance to host resistance. Furthermore, the Visium spatial transcriptomic analysis also demonstrates a delayed and sustained inflammatory response in K18-hACE2 mice versus a more rapid and resolving response in PWK x K18-hACE2, consistent with our interpretations of the initial cytokine/chemokine results in BAL. Together, these datasets significantly strengthen the conclusions.

MINOR comments:

Revise some of the missing labels in graphs, especially in supplemental figures.

Revise text, some sentence are unfinished (line 321)

Authors chose to use F1 progeny and this has its own limitations that can be introduced and discussed.

Authors may consider moving Fig. S4 to the main body.

Reply: Minor comments:

- 1) Missing labels, typing errors and incomplete sentences have been corrected throughout the figures and text.
- 2) We have included the limitations of using F1 mice in the (Discussion, lines 338-344).
- 3) To highlight the additional experiments and data included in the revised manuscript (Fig. 4-6), we elected to keep the heatmaps of the cytokine/chemokine levels in the supplemental materials.

Reviewer #3 (Remarks to the Authors):

Robertson et al. submitted a very interesting manuscript in which they investigated various outcomes of SARS-CoV-2 infections in several new mouse models that were generated from crosses of collaborative cross (CC) founder strains with one of the most widely used Covid-19 mouse model K18-hACE2.

Eight cohorts (n=6-7 per sex) of F1 offspring of the CC founder strains crossed with K18-hACE2 were investigated in a comparative approach regarding bodyweight loss, survival, viral replication kinetics, histopathology, and cytokine profiles. The result was a very interesting panel of different sensitivity or resistance to viral infection depending on which CC strain was mated to the classical model. In addition, sexual dimorphism was found in some of the CC x K18-hACE2 combinations. The different variants were then clustered to identify biomarker profiles associated with the different susceptibility to infection (antiviral response, inflammation).

Even after more than two years of the pandemic, the authors addressed a very relevant question, of which genetic setup could lead to such different severity of disease courses after SARS-CoV-2 infection in humans. The use of the so-called collaborative cross is an interesting complementary approach to elucidate this question because the CC may better represent genetic variability than other experimental approaches. Overall, the results show how valuable experimental mouse models can be for human disease, especially in the case of Covid-19.

My impression is that the study design is sound and all methods were applied correctly. Overall, the paper is very well written. I only had a little trouble with the long results section.

Reply: We thank the Reviewer for their positive comments. We have significantly strengthened the results section by shortening the extensive commentary on the cytokine/chemokine response and incorporating new and exciting data demonstrating transcriptional and IFN-I responses as they related to virus control and host susceptibility to disseminating infection.

However, I have two major concerns. (1) It is well known that the response of K18-hACE2 to SARS-CoV2 infection can be highly variable (which was also shown in this study). Thus, one has to ask how much variability was introduced by K18-hACE2 and what proportion by the different CC crossings.

Reply: We appreciate the Reviewer's comments. In the initial studies, we chose to use a sublethal dose to enable different outcomes without overwhelming innate resistance of a given genetic background. We suspect that the sublethal virus dose is a major contributor to the observed variability, and we have included it as a discussion point regarding the limitations of the study (Discussion, lines 338-344). We have also begun to address this experimentally by comparing the survival in K18-hACE2 and PWK x K18-hACE2 using a higher virus dose (10^4 pfu/mouse). As shown in Fig. S2a, all K18-hACE2 reached endpoint criteria by 5-7 dpi demonstrating that this dose was uniformly lethal in a sensitive strain. Although female PWK x K18-hACE2 showed greater sensitivity to the higher dose compared to 10^3 (60% lethality at 10^4 pfu compared to 20% at 10^3 pfu (Fig. 1b)), male PWK x K18-hACE2 maintained a resistant phenotype with 90-100% surviving infection (Fig. S2a). Thus, survival outcomes are tunable in some instances, based on the combination of virus dose and host genetics. We then used this more stringent approach in male mice to demonstrate the importance of the innate IFN-I response in controlling downstream inflammation and viral control in PWK x K18-hACE2 mice. This work attests to the biological relevance of our initial findings and further strengthens the manuscript.

(2) Unfortunately, it is not discussed at all whether the F1 offspring differ in their phenotype. This could be expected because the CC founders are already very different. Could an F1 phenotype have influenced the strength of the reaction to the virus infection? In conclusion, the authors present and suggest new models for further investigation, but in particular, the main idea of using genetically diverse mouse models is not explored in depth.

Because the CC founder strains and K18-hACE2 mice are entirely homozygous throughout their genomes, all F1 offspring within each cross (CC x K18-hACE2) will be heterozygous for all alleles that differ between the two parent strains. Moreover, F1 offspring within each cross are identical to each other, ruling out the possibility of genetics contributing to the observed variability of data within a given strain. However, F1 offspring from different CC founders x K18-hACE2 are indeed genetically different from each other and, according to our findings, phenotypically different in their response to SARS-CoV-2 infection. Compared to homozygous inbred mouse strains, the heterozygosity of F1 offspring can be a disadvantage because important phenotypes could be due to either or both alleles, making it difficult to map the genetic loci responsible for a given phenotype. However, K18-hACE2 mice were used as a common parent for all F1s and therefore, any differences observed in disease outcome is due to genes received from the unique parental strain (CC founders, BALB/cJ or DBA/2). Although genetic mapping using the K18-hACE2 models would be difficult, we think the strength of these models is that they can now be used to comprehensively characterize immunological and transcriptional responses to infection and their significance to viral control. In response to the Reviewers comment, we have added a better discussion of this point on lines 338-344.

REVIEWERS' COMMENTS

Reviewer #1 (Remarks to the Author):

The authors have satisfactorily addressed most of my concerns.

Reviewer #2 (Remarks to the Author):

Overall, if the intention of this study is to define and justify the use of the CC x K18 hACE2 mouse models to study host-dependent responses to SARS-CoV-2 infection, then authors addressed all my previous concerns. If we go into the mechanisms behind resistance to SARS-CoV-2 infection observed in some CC x K18hACE2 mice (e.g. PWK), of course, some questions remain, but this could be a follow study. As many possibilities, just as an example on the role of IFN-I signaling, it could be the case that SARS-CoV-2 can regulate the IFN-I production and signaling in K18 hACE2 transgenic mice but not in the PWK x K18 hACE2 mice driving resistance (e.g. able to block STAT1/2 nuclear translocation, a crucial step to activate the transcription of ISGs).

Reviewer #3 (Remarks to the Author):

I have carefully re-read the revised manuscript and the response letter. The criticisms I have raised have been addressed in the response letter and relevant passages have been added for further explanation. In addition, new data have been included in the manuscript which, in my view, further support the statements of the study. For this reason, I have no objections to the acceptance of the manuscript.

Response to Reviewers (NCOMMS-22-05743A):

Thank you to all 3 reviewers for their constructive comments that helped make the manuscript a more complete and interesting story.

Reviewer #2 (Remarks to the Author):

Overall, if the intention of this study is to define and justify the use of the CC x K18 hACE2 mouse models to study host-dependent responses to SARS-CoV-2 infection, then authors addressed all my previous concerns. If we go into the mechanisms behind resistance to SARS-CoV-2 infection observed in some CC x K18hACE2 mice (e.g. PWK), of course, some questions remain, but this could be a follow study. As many possibilities, just as an example on the role of IFN-I signaling, it could be the case that SARS-CoV-2 can regulate the IFN-I production and signaling in K18 hACE2 transgenic mice but not in the PWK x K18 hACE2 mice driving resistance (e.g. able to block STAT1/2 nuclear translocation, a crucial step to activate the transcription of ISGs).

Reply: We agree that our findings raise very interesting follow up questions including the one raised by Reviewer#2. We have added a sentence in the discussion to note this and other possibilities, and are excited to further explore these mouse models to experimentally address these critical questions.